# Comparison of three common nutritional screening tools with the new European Society for Clinical Nutrition and Metabolism (ESPEN) criteria for malnutrition among patients with geriatric gastrointestinal cancer: a prospective study in China

Xiao-Jun Ye,[1,2] Yan-Bin Ji,[1,3] Bing-Wei Ma,[1] Dong-Dong Huang,[1] Wei-Zhe Chen,[4] Zong-You Pan,[5] Xian Shen,[6] Cheng-Le Zhuang,[1] Zhen Yu[1]

X-JY and Y-BJ contributed equally.

For numbered affiliations see end of article.

**Correspondence to**
Dr. Cheng-Le Zhuang;
zhuangchengle@126.com and
Dr. Zhen Yu;
yuzhen0577@gmail.com

## ABSTRACT

**Objective** The aim of this study was to evaluate and compare three common nutritional screening tools with the new European Society for Clinical Nutrition and Metabolism (ESPEN) diagnostic criteria for malnutrition among elderly patients with gastrointestinal cancer.

**Research methodsandprocedures** Nutritional screening tools, including the Nutritional Risk Screening 2002 (NRS 2002), the Malnutrition Universal Screening Tool (MUST) and the Short Form of Mini Nutritional Assessment (MNA-SF), were applied to 255 patients with gastrointestinal cancer. We compared the diagnostic values of these tools for malnutrition, using the new ESPEN diagnostic criteria for malnutrition as the 'gold standards'.

**Results** According to the new ESPEN diagnostic criteria for malnutrition, 20% of the patients were diagnosed as malnourished. With the use of NRS 2002, 52.2% of the patients were found to be at high risk of malnutrition; with the use of MUST, 37.6% of the patients were found to be at moderate/high risk of malnutrition; and according to MNA-SF, 47.8% of the patients were found to be at nutritional risk. MUST was best correlated with the ESPEN diagnostic criteria (К=0.530, p<0.001) compared with NRS 2002 (К=0.312, p<0.001) and MNA-SF (К=0.380, p<0.001). The receiver operating characteristic curve of MUST had the highest area under the curve (AUC) compared with NRS 2002 and MNA-SF.

**Conclusions** Among the tools, MUST was found to perform the best in identifyingmalnourished elderly patients with gastrointestinal cancer distinguished by the new ESPEN diagnostic criteria for malnutrition. Nevertheless, further studies are needed to verify our findings.

**Trial registration number** ChiCTR-RRC-16009831; Pre-results.

## Strengths and limitations of this study

▶ To the best of our knowledge, this is the first study to evaluate the three screening tools in patients with specific geriatric gastrointestinal cancer.
▶ We compared the diagnostic value of the three screening tools using the new ESPEN diagnostic criteria for malnutrition as the 'gold standards'.
▶ The sample size is relatively small. However, this study was conducted in two centres with large surgical volumes. To a certain extent, it overcame the smallness of the sample size.

elderly patients has obviously enlarged. It is well known that the risk of cancer increases with age. More than half of the malignancies occur in people aged ≥65 years.[1 2] Gastrointestinal cancer is one of the most common malignancies in the elderly,[3] and surgical excision remains the most effective therapy for gastrointestinal cancers.[4] Although surgical techniques have improved significantly, there still exists a high frequency of complications and mortality in elderly patients with gastrointestinal cancer.[4–8] This is partly due to the high prevalence of malnutrition, which is a common and serious problem in elderly patients with cancer.[9–11] Therefore, it is important to evaluate the nutritional risk of patients with geriatric gastrointestinal cancer before surgery.

To accurately assess nutritional risk, it is important to choose an efficient nutritional screening tool. Although there are many widely used nutritional screening tools[12] such as the Nutritional Risk Screening 2002

## INTRODUCTION

As life expectancy and ages of the world population increase, the proportion of

(NRS 2002),[13] the Malnutrition Universal Screening Tool (MUST)[14] and the Short Form of Mini Nutritional Assessment (MNA-SF),[15] it has not been established which tool is most efficient and appropriate for nutritional screening in elderly patients with gastrointestinal cancers. Moreover, the lack of a universal definition of malnutrition may lead to inaccurate assessment and comparison of the nutritional screening tools.

Recently, a diagnostic criteria for malnutrition has been proposed by the European Society for Clinical Nutrition and Metabolism (ESPEN),[16] which had been validated by some studies.[17 18] This newly proposed criteria for defining malnutrition provides a reference standard for the evaluation and comparison of nutritional screening tools. Therefore, this study aims to evaluate the consistency of the three common nutritional screening tools with the new ESPEN diagnostic criteria for malnutrition and compare them among patients with geriatric gastrointestinal cancer.

## MATERIALS AND METHODS
### Patients
Between January 2016 and May 2017, 255 patients who underwent curative surgery for gastrointestinal cancer in two hospitals from Shanghai and Wenzhou were included in this study. The inclusion criteria were: (1) those who underwent elective curative surgery for gastrointestinal cancer; (2) those aged ≥70 years; (3) those who signed informed consent and agreed to participate in this study. The exclusion criteria were: (1) those who have undergone a palliative or emergency operation; (2) those aged <70 years; (3) those who cannot be assessed by NRS 2002, MUST and MNA-SF because of difficulty in data collection; (4) those who refused to take part in this study.

### Data collection
Patients' general and anthropometric data were collected. General data parameters were age, sex, diagnosis, morbidity, changes in appetite and physical activity. Anthropometric parameters included weight, height, unintentional weight loss and body mass index (BMI).

### Reference standard: the new ESPEN diagnostic criteria for malnutrition
According to the new ESPEN diagnostic criteria,[16] malnutrition was diagnosed when the patients met one of the following two options. Option 1 required BMI <18.5 kg/m$^2$. Option 2 required unintentional weight loss >10% in indefinite time or >5% over the last 3 months combined with reduced BMI (<20 kg/m$^2$ in patients younger than 70 years or <22 kg/m$^2$ in patients older than 70 years).

### Assessment of nutritional risk
The following nutritional screening tools were used for assessment: NRS 2002, MUST and MNA-SF.

NRS 2002 is proposed by ESPEN guidelines on the basis of analysis of controlled clinical trials.[13] It is designed to identify those who need nutritional support. This tool contains a severity of disease score, a nutritional score and an age score. Severity of disease score: one point for hip fracture, long-term haemodialysis, diabetes mellitus or chronic disease with acute complications; two points for major abdominal surgery, haematological malignancies, stroke or severe pneumonia; three points for head injury, bone marrow transplantation or patients in the intensive care unit with Acute Physiology and Chronic Health Evaluation (APACHE) >10. Nutritional score: one point for weight loss >5% in 3 months or food intake 50%–75% of the common condition; two points for weight loss >5% in 2 months or BMI 18.5–20.5 kg/m$^2$ with impaired general condition or food intake 25%–60% of the general condition; three points for weight loss >5% in 1 month or BMI less than 18.5 kg/m$^2$ with impaired general condition or food intake reduced by 25% compared with the normal condition. Age score: one point for age ≥70 years. Nutritional risk was assessed by summarising the disease severity score, nutritional score and age score. Patients with a total score <3 are at no or low risk, and those with a score ≥3 are at high risk.

MUST is developed to assess nutritional risk in adults.[14] It includes the following parameters: BMI, unintentional weight loss and any acute disease which compromises nutritional intake for more than 5 days. The three parameters rated as 0, 1 or 2 are as follows: BMI >20 kg/m$^2$=0, 18.5–20 kg/m$^2$=1, <18.5 kg/m$^2$=2; unintentional weight loss in the past 3–6 months <5%=0, 5%–10%=1, >10%=2; acute disease absent=0, present=2. Overall risk of malnutrition is assessed by adding all the points together; 0 is at low risk, score 1 is at medium risk, and score 2 is at high risk.

MNA-SF is the short form of MNA, and it is designed especially for the elderly. It contains six questions selected from MNA.[15] These questions are about BMI, recent weight loss, change of appetite, mobility, psychological stress and neuropsychological problems. Each question is rated from 0 to 2 or 3 and the total score of MNA-SF is 14. Patients with 12–14 points are at normal nutritional status. And patients with scores ≤11 are at risk of malnutrition.

### Statistical analysis
Statistical analysis was performed using SPSS software V.23.0 for Windows. Normally distributed continuous variables were expressed as mean values and SD, categorical variables were presented as absolute and relative frequencies. Independent t-test and Pearson's $\chi^2$ test (or Fisher's exact test) were applied for the appropriate comparison of variables. All reported p values were compared with a significance level of 5% based on two-sided tests. To determine diagnostic concordance between the three assessment tools and the new ESPEN diagnostic criteria for malnutrition, Cohen's κ statistic was calculated. The K coefficient reflects the consistency of qualitative variables.

**Table 1** Characteristics of all the patients

| Patient characteristics | Total (n=255) | GC (n=103) | CRC (n=152) | P value |
|---|---|---|---|---|
| Age* (years) | 76.5±4.8 | 76.1±4.6 | 76.8±4.9 | 0.318 |
| Sex† | | | | |
| Male | 160 (62.7%) | 78 (75.7%) | 82 (52.1%) | <0.001‡ |
| Female | 95 (37.3%) | 25 (24.3%) | 70 (47.9%) | |
| Height* (m) | 1.61±0.08 | 1.61±0.07 | 1.60±0.09 | 0.269 |
| Weight* (kg) | 59.20±10.73 | 59.67±10.26 | 58.90±11.07 | 0.597 |
| BMI* (kg/m$^2$) | 22.93±3.55 | 22.93±3.50 | 22.96±3.59 | 0.994 |
| BMI† (<18.5 kg/m$^2$) | 26 (10.2%) | 11 (10.7%) | 15 (9.9%) | 0.834 |
| Weight loss† (>5% in 3 months or >10% in indefinite time) | 74 (29.0%) | 31 (30.1%) | 43 (28.3%) | 0.755 |

P values were determined with the use of independent t-test and Pearson's $\chi^2$ test.
*Values expressed as mean ±SD.
†Values expressed as frequencies and percentages.
‡Statistical significance (p≤0.05).
BMI, body mass index; CRC, colorectal cancer; GC, gastric cancer.

K=1 means complete consistency between the variables. And if there is no consistency among the variables then K ≤0. Positive likelihood ratios and negative likelihood ratios were calculated for all the three tools.

Sensitivity and specificity values for the three nutritional screening tools with the new ESPEN diagnostic criteria for malnutrition were calculated. Receiver operating characteristic (ROC) curves of the three screening tools were also used to evaluate the ability to accurately distinguish malnourished patients. Area under the ROC curve (AUC) =0.5 indicates that a tool has no diagnostic value, AUC = 0.5–0.7 indicates a tool has a low diagnostic value, AUC =0.7–0.9 indicates that a tool has moderate diagnostic value, and AUC =0.9–1 means that a tool has a high diagnostic value.

## RESULTS

Two hundred and fifty-five patients were enrolled in this study (103 patients underwent gastric cancer surgery and 152 patients underwent colorectal cancer surgery). The characteristics of the sample are presented in table 1.

Table 2 lists the characteristics and anthropometric data of patients summarised and stratified according to nutritional status. There were no differences in age or sex between the two groups classified by the three screening tools and the new ESPEN criteria for malnutrition. However, BMI and weight loss (>5% in 3 months or >10% in indefinite time) differed between the groups.

Among the patients, the prevalence of malnutrition was 20.0% when determined by the ESPEN criteria. Among patients who underwent curative gastrectomy, the prevalence of malnutrition was 22.3% when determined by the new ESPEN criteria. And among patients who underwent colorectal surgery, the prevalence of malnutrition was 18.4% when determined by the new

ESPEN criteria. The classification of nutritional risk according to the three screening tools is shown in table 3.

Cross-tabulation of the results of the three tools and the classification of malnutrition according to the ESPEN consensus definition of malnutrition is given in table 4.

There was a difference in the consistencies between the nutritional screening tools and the new ESPEN diagnostic criteria for malnutrition. In all the patients, MUST and MNA had the same sensitivity (94.1%), and NRS 2002 had the lowest sensitivity (92.2%). Moreover MUST had the highest specificity (76.5%) compared with NRS 2002 (57.8%) and MNA-SF (63.7%). MUST had the highest positive predictive (50.0%) and negative predictive (98.1%) values.

In all the patients, MUST had the highest K value (K=0.530, p<0.001) compared with MNA-SF (K=0.380, p<0.001) and NRS 2002 (K=0.312, p<0.001). In the gastric group, MUST had the highest K value. In the colorectal group, MUST had a higher level of consistency (K=0.576, p<0.001) compared with the fair consistencies in NRS 2002 (K=0.243, p<0.001) and MNA-SF (K=0.361, p<0.001). Finally, the area under the curve (AUC) calculated by the ROC indicated that all three screening tools had a moderate level of diagnostic value to distinguish a malnourished patient (AUC of NRS 2002, MUST and MNA-SF were found to be 0.750, 0.853 and 0.789, respectively). Results are presented in detail in table 5.

## DISCUSSION

This study is the first to apply the new ESPEN diagnostic criteria for malnutrition specifically to the population of elderly patients with gastrointestinal cancer. Of the patients 20.0% were classified as malnourished according to this criteria. A previous study has investigated the prevalence of malnutrition diagnosed by the new ESPEN criteria in four diverse populations.[19] In that study, 0.5% of healthy

**Table 2** Characteristics and anthropometric data stratified by nutritional status of all the patients

| | ESPEN criteria | | | NRS 2002 | | | MUST | | | MNA-SF | | |
|---|---|---|---|---|---|---|---|---|---|---|---|---|
| | Not malnourished (n=204) | Malnourished (n=51) | P value | No or low risk (n=122) | High risk (n=133) | P value | Low risk (n=159) | Moderate /high risk (n=96) | P value | No risk (n=133) | Risk of malnutrition (n=122) | P value |
| Age* (years) | 76.3±4.8 | 77.4±4.9 | 0.148 | 75.9±4.5 | 77.0±5.0 | 0.059 | 76.1±4.6 | 77.1±5.1 | 0.097 | 76.0±4.6 | 77.0±5.0 | 0.102 |
| Sex† | | | | | | | | | | | | |
| Male | 124 (60.8%) | 36 (70.6%) | 0.195 | 73 (59.8%) | 87 (65.4%) | 0.357 | 101 (63.5%) | 59 (61.5%) | 0.741 | 90 (67.7%) | 70 (57.4%) | 0.089 |
| Female | 80 (39.2%) | 15 (29.4%) | | 49 (40.2%) | 46 (34.6%) | | 58 (36.5%) | 37 (38.5%) | | 43 (32.3%) | 52 (42.6%) | |
| Height* (m) | 1.60±0.08 | 1.63±0.08 | 0.046‡ | 1.60±0.08 | 1.61±0.09 | 0.155 | 1.60±0.08 | 1.61±0.09 | 0.511 | 1.61±0.08 | 1.60±0.08 | 0.112 |
| Weight* (kg) | 61.55±10.00 | 49.80±8.17 | <0.001‡ | 60.23±10.04 | 58.26±11.29 | 0.142 | 62.46±9.55 | 53.80±10.45 | <0.001‡ | 63.32±9.56 | 54.72±10.16 | <0.001‡ |
| BMI* (kg/m²) | 23.98±3.01 | 18.76±2.20 | <0.001‡ | 23.56±3.27 | 22.36±3.70 | 0.006‡ | 24.28±2.95 | 20.71±3.34 | <0.001‡ | 24.30±2.98 | 21.45±3.52 | <0.001‡ |
| BMI† (<18.5 kg/m²) | 1 (0.5%) | 25 (49.0%) | <0.001‡ | 4 (3.3%) | 22 (16.5%) | 0.001‡ | 1 (0.6%) | 25 (26.0%) | <0.001‡ | 1 (0.8%) | 25 (20.5%) | <0.001‡ |
| Weight loss† (>5% in 3 months or >10% in indefinite time) | 34 (16.7%) | 40 (78.4%) | <0.001‡ | 4 (3.3%) | 70 (52.6%) | <0.001‡ | 14 (8.8%) | 60 (62.5%) | <0.001‡ | 4 (3.0%) | 70 (57.4%) | <0.001‡ |

P values were determined with the use of independent t-test and Pearson's $\chi^2$ test (or Fisher's exact test, where appropriate).

*Values expressed as mean ±SD.

†Values expressed as frequencies and percentages.

‡Statistical significance (p≤0.05).

BMI, body mass index; ESPEN, European Society for Clinical Nutrition and Metabolism; MNA-SF, Short Form of Mini Nutritional Assessment; MUST, Malnutrition Universal Screening Tool; NRS 2002, Nutritional Risk Screening 2002.

**Table 3**  Classification of the risk of malnutrition with the ESPEN criteria and the three screening tools

| Risk of malnutrition | Total | | | GC (n=103) | | | CRC (n=152) | | |
|---|---|---|---|---|---|---|---|---|---|
| | NRS 2002 | MUST | MNA-SF | NRS 2002 | MUST | MNA-SF | NRS 2002 | MUST | MNA-SF |
| No/low | 47.8% (122/255) | 62.4% (159/255) | 52.2% (133/255) | 47.6% (49/103) | 54.4% (56/103) | 56.3% (58/103) | 48.0% (73/152) | 67.8% (103/152) | 49.3% (75/152) |
| Moderate /high | 52.2% (133/255) | 37.6% (96/255) | 47.8% (122/255) | 52.4% (54/103) | 45.6% (47/103) | 43.7% (45/103) | 52.0% (79/152) | 32.2% (49/152) | 50.7% (77/152) |

CRC, colorectal cancer; ESPEN, European Society for Clinical Nutrition and Metabolism; GC, gastric cancer; MNA-SF, Short Form of Mini Nutritional Assessment; MUST, Malnutrition Universal Screening Tool; NRS 2002, Nutritional Risk Screening 2002.

elderly individuals and 6% of geriatric outpatients were identified as malnourished. These values are significantly lower compared with those of our study, which indicates that patients with gastrointestinal cancer might have a higher prevalence of malnutrition. The difference in the malnutrition rates also emphasises the need to assess nutritional risk of hospitalised elderly patients with gastrointestinal cancer.

In the present study, 52.2% and 37.6% of the patients were found to be at moderate or high risk of malnutrition according to NRS 2002 and MUST, respectively. With MNA-SF, 47.8% of the patients were found to be at risk of malnutrition. The different prevalences of the risk of malnutrition could be a result of the differences between the nutritional screening tools. In our study, MUST had the greatest K value compared with NRS 2002 and MNA-SF. It showed that a greater proportion of elderly patients with gastrointestinal cancer, who were found to be at moderate or high risk of malnutrition with MUST, could be identified as malnourished according to the ESPEN diagnostic criteria. In other words, MUST is best at detecting the specific malnourished individuals diagnosed by the new ESPEN criteria, compared with NRS 2002 and MNA-SF. Furthermore, MUST was found to have the greatest AUC compared with NRS 2002 and MNA-SF, in our study.

Many previous studies compared the three nutritional screening tools in specific populations. Poulia et al evaluated the efficacy of six nutritional screening tools in the elderly.[20] In Poulia's study, NRS 2002 was found to overestimate nutritional risk, MNA-SF was proven to have great validity, and MUST was found to have the best validity and the greatest consistency. Another study by Myoungha et al[21] evaluated five nutritional screening tools, and

suggested that MNA-SF overestimated nutritional risk in the elderly, and NRS 2002 performed better than MNA-SF. However, MUST was also found to be the most efficient and useful screening tool in this study. Both the previous studies compared screening tools with a combined index suggested by Pablo et al,[22] and they confirmed our results for the best performance of MUST compared with NRS 2002 and MNA-SF. Donini et al developed a study for nutritional evaluation of elderly nursing home residents[23] and found that MNA-SF presented a higher predictive value compared with NRS 2002 and MUST. However, in the study by Donini et al, MNA was taken as the reference standard, which might induce an underpowered result. A previous study used the new ESPEN diagnostic criteria for malnutrition as a reference standard to compare nutritional screening tools.[24] However, MNA-SF was not included in this study, and the participants were not merely the elderly.

In our study, the results of the comparison between NRS 2002, MUST and MNA-SF showed different efficiencies of the three screening tools. There might be an explanation for these differences. The original designs of the three nutritional screening tools were different. NRS 2002 was developed to determine who needs nutritional support and it might identify an increased number of patients to be at high risk of malnutrition. According to the diagnosis criteria of NRS 2002, one score was added to patients aged ≥70 years, it might contribute to the prevalence of patients at nutritional risk diagnosed by NRS 2002. However, MUST was a screening tool for identifying adults who are at risk of malnutrition. MNA-SF was developed as a quick and easy nutritional screening tool and was used for primary screening before further assessment.

**Table 4**  Cross-tabulation of the results of the three screening tools and the classification of malnutrition according to the ESPEN consensus definition of malnutrition

| | NRS 2002 | | MUST | | MNA-SF | |
|---|---|---|---|---|---|---|
| | No/low risk | Moderate/high risk | Low risk | Moderate/high risk | No risk | Risk |
| ESPEN criteria | | | | | | |
| Not malnourished | 118 | 86 | 156 | 48 | 130 | 74 |
| Malnourished | 4 | 47 | 3 | 48 | 3 | 48 |

MNA-SF, Short Form of Mini Nutritional Assessment; ESPEN, European Society for Clinical Nutrition and Metabolism; MUST, Malnutrition Universal Screening Tool; NRS 2002, Nutritional Risk Screening 2002.

**Table 5** Statistical evaluation of the malnutrition screening tools compared with the diagnostic criteria of the ESPEN consensus

| | Total | | | GC | | | CRC | | |
|---|---|---|---|---|---|---|---|---|---|
| | NRS 2002 | MUST | MNA-SF | NRS 2002 | MUST | MNA-SF | NRS 2002 | MUST | MNA-SF |
| Sensitivity (%) | 92.2 | 94.1 | 94.1 | 100.0 | 95.7 | 87.0 | 85.7 | 92.9 | 100.0 |
| Specificity (%) | 57.8 | 76.5 | 63.7 | 61.3 | 68.8 | 68.8 | 55.6 | 81.5 | 60.5 |
| Positive predictive value (%) | 35.3 | 50.0 | 39.3 | 42.6 | 46.8 | 44.4 | 30.4 | 53.1 | 36.4 |
| Negative predictive value (%) | 96.7 | 98.1 | 97.7 | 100.0 | 98.2 | 94.8 | 94.5 | 98.1 | 100.0 |
| Positive likelihood ratio (LR+) | 2.18 | 4.00 | 2.59 | 2.61 | 3.06 | 2.78 | 1.93 | 5.02 | 2.53 |
| Negative likelihood ratio (LR–) | 0.13 | 0.08 | 0.09 | 0.00 | 0.06 | 0.19 | 0.26 | 0.09 | 0.00 |
| K* value (p) | 0.312 (<0.001) | 0.530 (<0.001) | 0.380 (<0.001) | 0.414 (<0.001) | 0.469 (<0.001) | 0.415 (<0.001) | 0.243 (<0.001) | 0.576 (<0.001) | 0.361 (<0.001) |
| AUC | 0.750 | 0.853 | 0.789 | 0.806 | 0.822 | 0.779 | 0.707 | 0.872 | 0.802 |

*K value derived from Cohen's κ statistics.
AUC, area under the curve from ROC; CRC, colorectal cancer; ESPEN, European Society for Clinical Nutrition and Metabolism; GC, gastric cancer; MNA-SF, Short Form of Mini Nutritional Assessment; MUST, Malnutrition Universal Screening Tool; NRS 2002, Nutritional Risk Screening 2002.

The different results between our study and the others' can be attributed to the different populations and reference standards, similar to what Ma *et al* have mentioned in their review.[25]

Moreover, in our study, we evaluated the adequacy of the nutritional screening tools in patients with gastric cancer and patients with colorectal cancer, respectively. In the gastric cancer population, NRS 2002, MUST and MNA-SF all had the same moderate level of consistency with the new ESPEN diagnostic criteria for malnutrition (K=0.414 for NRS 2002, 0.469 for MUST and 0.415 for MNA-SF, respectively). While in the colorectal cancer population, MUST had the highest level of consistency (K=0.576) compared with the fair level of consistency of NRS 2002 (K=0.243) and MNA-SF (K=0.361). Based on the result, we concluded that MUST could perform best specifically in patients with colorectal cancer. Further studies are needed to confirm this.

It is important to improve the nutritional status of elderly patients with gastrointestinal cancer if they are malnourished or at risk of malnutrition. As the first step to identify patients who are malnourished or at risk of malnutrition, nutritional screening should be reliable and easy to perform. To the best of our knowledge, this is the first study comparing the three nutritional screening tools in the specific geriatric gastrointestinal cancer population. Our study suggests that MUST is the best choice among the three common nutritional screening tools. Both, patients and surgeons will benefit from the accuracy and simplicity of MUST. Furthermore, in our study, though there were some differences in sensitivity and specificity, the three tools were found to have the same level of consistency in the gastric cancer population. While in the colorectal cancer population, MUST had the highest

level of consistency compared with the others (K=0.576). MUST also had a significantly higher specificity and positive predictive value, and greater AUC. This indicates that although we can choose any one of the three nutritional screening tools when assessing nutritional risk among elderly patients with gastric cancer, when assessing nutritional risk among elderly patients with colorectal cancer, MUST should be chosen for highest accuracy.

Although ESPEN guidelines promote NRS 2002 as the tool to screen hospitalised patients, the results of this study found MUST to be better for screening patients with gastrointestinal cancer. This could be due to several aspects. MUST was developed as a valid tool to identify nutritional risk of a specific patient population, and it is a tool uniquely designed for screening of malnutrition.[26] Therefore, MUST could perform best according to the ESPEN diagnostic criteria for malnutrition. Moreover, MUST has straightforward and objective questions, making it easier to use. It can be a useful nutritional screening tool when there is no redundant time and no professional medical staff.

It is well known that malnourished patients or patients at risk of malnutrition would have a poor clinical outcome. Therefore, it is important to improve the nutritional status of these patients. The indices of these three screening tools have common characteristics. With the results of our study, the parameters of MUST suggest that we could manage malnutrition by improving BMI, avoiding weight loss and curing the acute disease. In fact, this is partly similar to the parameters of the EPSEN diagnostic criteria, NRS 2002 and MNA-SF. It means that if the result of MUST was improved, the consequences of the EPSEN diagnostic criteria and other screening tools would also be improved. Further studies should be

developed to investigate implications of the intervention for malnutrition.

The present study has some limitations. First, the sample size is relatively small. However, this study was conducted at two centres with large surgical volumes. So we believe that the data in our study are more representative. To a certain extent, it overcame the smallness of the sample size. Second, according to the new ESPEN diagnostic criteria for malnutrition, malnutrition can also be diagnosed by unintentional weight loss combined with reduced fat-free mass index (FFMI). So another limitation of our study is the lack of data for FFMI. However, the measurement of FFMI requires specific equipment and extra costs. Moreover, Trummer $et$ $al$ found that low BMI and low FFMI were closely correlated.[27] In that study, FFMI less than $17 \text{kg/m}^2$ for men and less than $15 \text{kg/m}^2$ for women were roughly equivalent to BMI less than $18.5 \text{kg/m}^2$ after determining the fat-free mass (FFM) levels.

## CONCLUSIONS

To our knowledge, this is the first study to compare the three malnutrition screening tools (NRS 2002, MUST and MNA-SF) with the new ESPEN diagnostic criteria for malnutrition, and it is also the first study to evaluate the three screening tools specifically in patients with geriatric gastrointestinal cancer . The prevalence of malnutrition was 20.0% with the ESPEN diagnostic criteria for malnutrition for patients with gastrointestinal cancer in the present study. MUST was found to perform the best to identify the malnourished elderly patients with gastrointestinal cancer distinguished by the new ESPEN diagnostic criteria for malnutrition. Nevertheless, further studies are needed to verify our findings.

**Author affiliations**
[1]Department of Gastrointestinal Surgery, Shanghai Tenth People's Hospital Affiliated to Tongji University, Shanghai, China
[2]Department of Medicine, Medical College of Soochow University, Suzhou, China
[3]Department of Gastrointestinal Surgery, Shanghai Clinical Institution, Anhui Medical University, Shanghai, China
[4]The First Affiliated Hospital, Wenzhou Medical University, Department of Gastrointestinal Surgery, Wenzhou, China
[5]Department of Sports Medicine, School of Medicine, Zhejiang University, Hangzhou, China
[6]Department of Gastrointestinal Surgery, The Second Affiliated Hospital, Wenzhou Medical University, Wenzhou, China

**Contributors** X-JY and Y-BJ are joint first authors. ZY, C-LZ and XS designed the study. B-WM and W-ZC collected the data. D-DH and Z-YP analysed the data. X-JY and Y-BJ drafted the manuscript. ZY and C-LZ approved the final version of the manuscript. All authors have read and approved the final manuscript.

**Funding** This work was funded by the Science and Technology Commission of Shanghai Municipality (grant number 16411954200); and Shanghai Municipal Commission of Health and Family Planning [grant number 201640013].

**Competing interests** None declared.

**Patient consent** Obtained.

**Ethics approval** Ethics committees of the Tenth Affiliated Hospital of Tongji University and the First Affiliated Hospital of Wenzhou Medical University.

**Provenance and peer review** Not commissioned; externally peer reviewed.

**Data sharing statement** No additional unpublished data are available.

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
