## [Reviewer comments · BMJ Open]

ARTICLE DETAILS

TITLE (PROVISIONAL)	Comparison of three common nutritional screening tools with the new European Society for Clinical Nutrition and Metabolism (ESPEN) criteria for malnutrition among geriatric gastrointestinal cancer patients: a prospective study in China
AUTHORS	Ye, Xiao-Jun; Ji, Yan-Bin; Ma, Bing-Wei; Huang, Dong-Dong; Chen, Wei-Zhe; Pan, Zong-You; Shen, Xian; Zhuang, Cheng-Le; Yu, Zhen

VERSION 1 – REVIEW

REVIEWER	Carla Alberici Pastore Federal University of Pelotas, Brazil
REVIEW RETURNED	08-Oct-2017

GENERAL COMMENTS	Excellent paper, over a very important issue. The research and the manuscript preparation was careful and well done. The only suggestion I have is the correction of a typing error, in Page 10 of 25 of the manuscript pdf proof: "Area under the ROC curer, changing "curer" for "curve". Congrats to the authors and I look forward to read the published version soon.
--

REVIEWER	Paweł Kabata Department of Surgical Oncology, Medical University of Gdańsk, Poland
REVIEW RETURNED	16-Dec-2017

GENERAL COMMENTS	1. Page 8, line 7-9: NRS2002 is a tool to rather assess who needs nutritional support than who will benefit from this support.2. It is very surprising to see comparable malnutrition rates for patients with gastric and colorectal cancers. Usually upper GI cancers are more metabolically engaging. What was the cancer staging in these groups?? Probably this could be discussed.3. Table 3 should be redesigned as it is misleading, confusing and hard to understand. Putting malnutrition diagnosis according to ESPEN guidelines in the column section together with malnutrition risk screening tools and crossing them with malnutrition risk makes it look like it's one of the tools.4. What's the point of doubling the data in table 3 and the text above it.5. Authors should discuss their results against ESPEN Guidelines on nutritional screening and nutrition in cancer, especially that it used to promote NRS2002 as a tool to screen hospitalized patients.
---

REVIEWER	Cecilia Gavazzi
-----------------	-----------------

	Fondazione IRCCS National Cancer Institute Italy
REVIEW RETURNED	22-Dec-2017

GENERAL COMMENTS	The study is well conducted and compare the use of nutritional screening tools versus a new diagnostic criteria for malnutrition in two specific population. □ In table 1 characteristics in gastric and colorectal cancer group are presented. Both group have the same percentage of weight loss , this data is not in line other published data, were gastric cancer has a greater prevalence of weight loss in patients candidated to surgery. Wan-H Hu,Nutr J. 2015; □ In the evaluation of nutritional risk with the NRS, 1 score is added to patients aged ≥ 70years, therefore this is probably the reason of higher prevalence of patients at risk with this tool. Analysis of this point should be included in the results and discussion. □ Authors suggest that MUST is the best choice for the highest accuracy in colorectal cancer, this result should better analyzed in the discussion □ Discussion should include more concern related to the implication of finding an old patients candidate to surgery at risk of malnutrition or malnourished: i.e. which intervention it is then suggested: similar or different according to the risk of malnutrition, the different score or the diagnosis of malnutrition.
---

VERSION 1 – AUTHOR RESPONSE

List of Actions

- 1: The title was revised as the preferred format for the journal.
- 2: We fully spelled out the abbreviations.
- 3: We worked to improve the quality of the English.
- 4: We added the registration number in page 2.
- 5: We revised the statement of NRS 2002 in page 8 and page 18.
- 6: We revised the Table 3 and the text above it.
- 7: We added some analysis of MUST in colorectal cancer in page 19.
- 7: A discussion of NRS 2002 was added in page 18.
- 8: 2 paragraphs of discussion were added in page 20.
- 9: A reference was added in page 26.

Dear Editors,

Thanks for your letter concerning our manuscript. We are very sorry for our deficiency and we have studied the comments carefully to make correction which we hope to meet with approval. The manuscript has been revised by using the track changes mode in MS Word.

We have revised the title as the appropriate format and all abbreviations are fully spelled out. Also, we have complete the STARD check-list.

We tried our best to improve the manuscript and made some changes marked in the revised paper. Appreciate for your warm work earnestly and hope that the correction will meet the approval. Once again, thank you very much for your comments and suggestions.

Responses to Reviewers

To reviewer 1:

Thank you for pointing this out. We are very sorry for our negligence of the typing error and we have corrected it. Once again, thanks for your warn work earnestly.

To reviewer 2:

Thanks for your comments concerning our manuscript. Those comments are all valuable and very helpful for revising and improving our paper. The responses are as following:

1. Page 8, line 7-9: NRS2002 is a tool to rather assess who needs nutritional support than who will benefit from this support.

We have revised the statement of NRS 2002 as that it is a tool to assess who needs nutritional support in page 8 and page 18.

2. It is very surprising to see comparable malnutrition rates for patients with gastric and colorectal cancers. Usually upper GI cancers are more metabolically engaging. What was the cancer staging in these groups? Probably this could be discussed.

Thanks for reviewer's good comments. For the objective of this paper, we did not compare the malnutrition rates between patients with gastric and colorectal cancers. We evaluated the value of the three common nutritional screening tools to find which is the best for the total gastrointestinal cancer patients and then evaluated them for the gastric and colorectal cancer patients separately. With reviewer's comments, we realized that we could evaluated the screening tools in a more precise group. In the future, we should do it as the sample size is big enough. Also, we just investigated the accuracy of the nutritional screening tools simply in all cancer patients without the cancer staging, but the staging of the patients in our study is consecutive and practical. So the results can reflect the practical situation and could be significant. However, with the comments of reviewer, we should develop further studies to discuss the cancer staging of the gastric and colorectal cancer patients and it could be more accurate.

Thank you for the valuable comments again.

3. Table 3 should be redesigned as it is misleading, confusing and hard to understand. Putting malnutrition diagnosis according to ESPEN guidelines in the column section together with malnutrition risk screening tools and crossing them with malnutrition risk makes it look like it's one of the tools.

We have revised the Table 3 to make it clear and easy to understand.

4. What's the point of doubling the data in table 3 and the text above it.

We have deleted the repeating data. Thanks for the comments.

5. Authors should discuss their results against ESPEN Guidelines on nutritional screening and nutrition in cancer, especially that it used to promote NRS2002 as a tool to screen hospitalized patients.

We have added a paragraph of discussion for it in page 20. This might owe to several aspects. MUST was developed as a valid tool to identify nutritional risk of specific patients population, and it is the unique tool that was designed for screening of malnutrition. Therefore, MUST might perform best according to the ESPEN diagnostic criteria for malnutrition. Moreover, MUST has straight forward and objective questions, with which is easier to be performed. It can be a useful nutritional screening tool when there is no redundant time and no professional medical staff.

Special thanks to you for your good comments.

To reviewer 3:

Thank reviewer's comments and suggestions for our manuscript. The comments are valuable and help us to improve our paper. The responses are as following:

1. In table 1 characteristics in gastric and colorectal cancer group are presented. Both group have the same percentage of weight loss, this data is not in line other published data, were gastric cancer has a greater prevalence of weight loss in patients candidated to surgery. Wan-H Hu, Nutr J. 2015

Thanks for reviewer's good comments. We think this might owe to the heterogeneity of the crowds.

2. In the evaluation of nutritional risk with the NRS, 1 score is added to patients aged ≥ 70 years, therefore this is probably the reason of higher prevalence of patients at risk with this tool. Analysis of this point should be included in the results and discussion.

We have added this point and discuss it in page 18.

3. Authors suggest that MUST is the best choice for the highest accuracy in colorectal cancer, this result should better analyzed in the discussion.

We have added some analysis of MUST in colorectal cancer in page 20. We suggest MUST as the best choice for its highest level of consistency and the significant highest specificity, positive predictive value, and the greatest AUC.

4. Discussion should include more concern related to the implication of finding an old patients candidate to surgery at risk of malnutrition or malnourished: i.e. which intervention it is then suggested: similar or different according to the risk of malnutrition, the different score or the diagnosis of malnutrition.

We have added two paragraphs of discussion for it in page 20. We are very sorry for our deficiency. MUST was developed as a valid tool to identify nutritional risk of specific patients population, and it is the unique tool that was designed for screening of malnutrition. Therefore, MUST might perform best according to the ESPEN diagnostic criteria for malnutrition. Moreover, MUST has straight forward and objective questions, with which is easier to be performed. It can be a useful nutritional screening tool when there is no redundant time and no professional medical staff.

With the results of our study, the parameters of MUST suggest that we could intervene malnutrition by improving BMI, avoiding weight loss and curing the acute disease. However, in fact, this is partly similar with the parameters of EPSEN diagnostic criteria, NRS 2002 and MNA-SF. It means that if the result of MUST was improved, the consequence of EPSEN diagnostic criteria and other screening tools would also be improved.

We realized that further studies should be developed to investigate implications of the intervention for malnutrition.

Special thanks to you for your good comments.

VERSION 2 – REVIEW

REVIEWER	Paweł Kabata Department of Surgical Oncology, Medical University of Gdansk, Poland
REVIEW RETURNED	21-Feb-2018
GENERAL COMMENTS	The authors addressed my concerns adequately. I have no further comments.